# Maximal Intensity Exercise Induces Adipokine Secretion and Disrupts Prooxidant–Antioxidant Balance in Young Men with Different Body Composition

**DOI:** 10.3390/ijms26010350

**Published:** 2025-01-03

**Authors:** Magdalena Wiecek, Mateusz Mardyla, Jadwiga Szymura, Malgorzata Kantorowicz, Justyna Kusmierczyk, Marcin Maciejczyk, Zbigniew Szygula

**Affiliations:** 1Department of Physiology and Biochemistry, Institute of Biomedical Sciences, Faculty of Physical Education and Sport, University of Physical Education in Kraków, 31-571 Kraków, Poland; magdalena.wiecek@awf.krakow.pl (M.W.); justyna.kusmierczyk@awf.krakow.pl (J.K.); marcin.maciejczyk@awf.krakow.pl (M.M.); 2Department of Clinical Rehabilitation, Faculty of Motor Rehabilitation, University of Physical Education in Kraków, 31-571 Kraków, Poland; jadwiga.szymura@awf.krakow.pl; 3Medical Institute, Academy of Applied Sciences in Nowy Targ, 34-400 Nowy Targ, Poland; malgorzata.kantorowicz@ans-nt.edu.pl; 4Department of Sports Medicine and Human Nutrition, Institute of Biomedical Sciences, Faculty of Physical Education and Sport, University of Physical Education in Kraków, 31-571 Kraków, Poland; zbigniew.szygula@awf.krakow.pl

**Keywords:** exercise, adipokines, asprosin, oxidative stress, antioxidant enzymes, body composition

## Abstract

Maximal physical effort induces a disturbance in the body’s energy homeostasis and causes oxidative stress. The aim of the study was to determine whether prooxidant–antioxidant balance disturbances and the secretion of adipokines regulating metabolism, induced by maximal intensity exercise, are dependent on body composition in young, healthy, non-obese individuals. We determined changes in the concentration of advanced protein oxidation products (AOPP), markers of oxidative damage to nucleic acids (DNA/RNA/ox), and lipid peroxidation (LPO); catalase (CAT), superoxide dismutase (SOD), and glutathione peroxidase (GPx) activity, as well as concentrations of visfatin, leptin, resistin, adiponectin, asprosin, and irisin in the blood before and after maximal intensity exercise in men with above-average muscle mass (NFAT-HLBM), above-average fat mass (HFAT-NLBM), and with average body composition (NFAT-NLBM). We corrected the post-exercise results for the percentage change in plasma volume. In all groups after exercise, there was an increase in LPO and resistin. In HFAT-NLBM, additionally, an increase in CAT and a decrease in SOD activity were noted, and in NFAT-NLBM, an increase in visfatin concentration was observed. In our study, the effect was demonstrated of a maximal effort on six (LPO, CAT, SOD, visfatin, resistin, and asprosin) of the twelve parameters investigated, while the effect of body composition on all parameters investigated was insignificant. Maximal intensity aerobic exercise induces secretion of resistin and damages lipids regardless of the exercising subjects’ body composition. Large fat tissue content predisposes to exercise-induced disorders in the activity of antioxidant enzymes. We have also shown that it is necessary to consider changes in blood plasma volume in the assessment of post-exercise biochemical marker levels.

## 1. Introduction

Physical effort is a factor causing homeostasis disturbances, which result from the need to cover the increased demand for energy [1]. As a result of a single exercise bout in untrained individuals, an intensity- and time-dependent increase in the concentration of oxidized low-density lipoprotein, uric acid, antioxidant vitamins, and the level of oxidative stress index (OSI), as well as a decrease in the ratio of reduced to oxidized glutathione concentration (GSH/GSSG), might be observed in the blood [2,3,4,5]. In the early phase following exercise, the activity of antioxidant enzymes can be temporally decreased (catalase—CAT, superoxide dismutase—SOD, glutathione peroxidase—GPx), which indicates post-exercise systemic oxidative stress (OS) [6]. Reactive oxygen species (ROS) can act as signals of cellular adaptation and stimulate processes such as mitochondrial biogenesis, muscle hypertrophy, or activation of signaling pathways related to immunity and regeneration [7,8,9]. ROS can affect the secretion of adipokines via the modification of signaling pathways in adipocytes and muscle tissue [10,11,12]. Adipokines, myokines, and ROS participate in the regulation of metabolic processes induced by physical exercise [10]. Decreased leptin and resistin levels, as well as increased adiponectin and irisin levels, have been reported in response to acute aerobic and resistance exercise [10]. The adverse effects of excess ROS and its prolonged impact on cellular structures may be oxidative damage to lipids (LPO), DNA and RNA (DNA/RNA/ox), and proteins (AOPP), as well as dysregulation regarding the secretion of adipokines involved in the regulation of energy substrate availability [8,13].

The results of studies conducted so far, mainly on obese individuals, show positive correlations between the levels of OS markers, pro-inflammatory adipokines, and BMI in men and women of different ages [10,13]. However, increased BMI values may result not only from excess adipose tissue but also from increased skeletal muscle tissue [14,15]. Therefore, both adipose and muscle tissue may significantly contribute to systemic OS. In the context of maximal-intensity exercise, the role of ROS and myokines in inducing the secretion of adipokines into the bloodstream should be considered [8,10]. However, most of the studies concern obese populations and there is a lack of comprehensive consideration regarding various hormones released by adipose tissue in response to a single maximal exercise effort, in which the effects of OS in healthy young people are taken into account.

The aim of the present study is to determine whether maximal-intensity exercise induces similar changes in the concentration of adipokines and markers of the prooxidant–antioxidant balance among individuals with increased adipose tissue and in those with increased muscle tissue, which are more intense than in people with average body composition. In the current trial, changes were assessed in the concentration of adipokines, i.e., visfatin, leptin, resistin, adiponectin, asprosin, and irisin, markers of oxidative damage to proteins, lipids, and nucleic acids, as well as the activity of antioxidant enzymes following maximal intensity exercise in young men with above-average adipose tissue content (HFAT-NLBM group) and above-average muscle tissue content (NFAT-HLBM group). This was performed in comparison to individuals with average body composition (NFAT-NLBM group). To eliminate the influence of blood hemoconcentration on the falsification of the results, in our study, the post-exercise levels of biochemical markers were corrected by taking the percentage changes in plasma volume (%dPV) induced by intense physical exercise into account.

The following research hypotheses were posed: (1) maximal intensity exercise causes greater oxidative damage to (a) lipids, (b) proteins, and (c) nucleic acids, as well as (d) greater activation of antioxidant enzymes in people with above-average body fat and in people with above-average muscle mass in comparison to individuals with average body composition; (2) maximal intensity exercise causes (a) greater up-regulation of adiponectin and asprosin secretion and (b) greater down-regulation of leptin, resistin, and visfatin secretion in individuals with increased body fat compared to individuals with increased muscle mass and average body composition; whereas (c) the increase in irisin concentration is greater in individuals with increased muscle mass compared to those with increased body fat and average body composition.

## 2. Results

### 2.1. Participant Characteristics

The study groups were similar in age (*p* = 0.89), but differed significantly in terms of body mass (BM, *p* < 0.01), body mass index (BMI, *p* < 0.01), percentage of body fat (%FAT, *p* < 0.01), lean body mass (LBM, *p* < 0.01), and maximal oxygen uptake per minute in relation to body mass (VO_2_max × BM^−1^, *p* < 0.01). The age, body composition, and physical performance characteristics of the participants are presented in Table 1.

Post hoc analysis showed that the NFAT-HLBM group demonstrated significantly higher BM (*p* < 0.01, Dunn’s test), BMI (*p* < 0.01, Tukey’s HSD test), and LBM (*p* < 0.01, Tukey’s HSD test) and significantly lower VO_2_max × BM^−1^ (*p* = 0.03, Tukey’s HSD test) compared to the NFAT-NLBM group. The HFAT-NLBM group exhibited significantly higher BM (*p* < 0.01, Dunn’s test), BMI (*p* < 0.01, Tukey’s HSD test), and %FAT (*p* < 0.01, Tukey’s HSD test) and lower VO_2_max × BM^−1^ (*p* < 0.01, Tukey’s HSD test) compared to the NFAT-NLBM group. At the same time, the HFAT-NLBM group was characterized by significantly higher %FAT (*p* < 0.01, Tukey’s HSD test) and lower LBM (*p* < 0.01, Tukey’s HSD test) compared to the NFAT-HLBM group (Table 1).

In Table 2, the results regarding the medical qualifications of the participants are presented. There were no statistically significant differences between the groups in the content of morphotic blood elements, hemoglobin concentrations, or hematocrit values (*p* > 0.05). Fasting glucose concentrations, percentages of glycated hemoglobin, lipid profile, and C-reactive protein concentrations did not differ significantly between the groups (*p* > 0.05). Only the erythrocyte sedimentation rates (ESR) differed significantly between the groups (*p* = 0.03). Post hoc analysis (Dunn’s test) showed that the ESR was comparable in all groups (*p* > 0.05).

The results of somatic measurements and medical qualification confirm that the participants from each group were healthy, without inflammation, and not obese.

### 2.2. Plasma Volume Changes Following Maximal Intensity Exercise

As a result of maximal intensity exercise, plasma volume decreased by 7.25 ± 4.41% in the NFAT-NHLBM group as well as by 10.61 ± 4.83% and 10.86 ± 6.97% in the NFAT-HLBM and HFAT-NLBM groups, respectively. %dPV were comparable in all groups (*p* = 0.29, H = 2.51, Kruskal–Wallis test).

### 2.3. Effect of Maximal Intensity Exercise on Biochemical Markers

Table 3 presents the results of two-way repeated measures analysis of variance (two-way RM ANOVA), showing the influence of the main factors, i.e., BODY COMPOSITION, EXERCISE and the interaction of BODY COMPOSITION × EXERCISE, on the level of the analyzed biochemical markers in blood:

(1) Without correction for %dPV of post-exercise values (six groups results: NFAT-NLBM T0, NFAT-NLBM T1, NFAT-HLBM T0, NFAT-HLBM T1, HFAT-NLBM T0, HFAT-NLBM T1);

(2) With correction for %dPV of post-exercise values (six groups results: NFAT-NLBM T0, NFAT-NLBM T1_PV_, NFAT-HLBM T0, NFAT-HLBM T1_PV_, HFAT-NLBM T0, HFAT-NLBM T1_PV_).

The numerical results (mean, standard error—SE and standard deviation—SD) for individual biochemical markers obtained at all measurement points, together with the indication of statistically significant differences between post-exercise and pre-exercise results in each group (Tuckey’s HSD post hoc test independently for (1) and (2) two-way RM ANOVA; *p* < 0.05 for T1 vs. T0 and *p* < 0.05 for T1PV vs. T0) are presented graphically in Figure 1 and Figure 2 in the manuscript. Additionally, Appendix A present the mean values and SD for individual variables (Appendix A) and post-exercise changes (mean and 95% confidence interval—95% CI) in the analyzed variables, together with the results (*p*-value) of the post hoc analysis (Tuckey’s HSD test) for changes induced by exercise in each of the studied groups (Appendix A).

The inclusion of %dPV in the analysis of values obtained after exercise resulted in different results of the statistical analysis of seven of the twelve analyzed biochemical markers (Table 3). In the case of AOPP, GPx, leptin, adiponectin, and irisin, a significant effect of the EXERCISE factor on the obtained results was found only when %dPV for the values after exercise was not included (results of analysis (1): %dPV uncorrected). In contrast, in the case of SOD and asprosin, a significant effect of the EXERCISE factor on the obtained results was found only when correction for %dPV was applied (results of analysis (2): %dPV corrected).

The description of the results presented in the manuscript includes only values corrected for %dPV.

#### 2.3.1. Prooxidant–Antioxidant Balance

Two-way RM ANOVA showed that there was no significant effect of the BODY COMPOSITION factor on the concentration of AOPP, DNA/RNA/ox, LPO, or the activity of CAT, SOD, or GPx (*p* > 0.05). There was a significant effect of the EXERCISE factor on the concentration of LPO (*p* < 0.01), as well as on the change in the activity of CAT (*p* < 0.01) and SOD (*p* < 0.01). There was no significant interaction of the BODY COMPOSITION × EXERCISE factors on the level of the analyzed prooxidant–antioxidant balance indicators (*p* > 0.05) (Table 3).

Post hoc analysis (Tukey’s HSD test) showed that all groups exhibited a significant increase in the concentration of LPO in NFAT-NLBM (*p* = 0.03), NFAT-HLBM (*p* = 0.049), and HFAT-NLBM (*p* = 0.01), respectively. In the HFAT-NLBM group, CAT activity increased (*p* = 0.048), while SOD activity decreased significantly (*p* = 0.049) (Figure 1, Appendix A).

#### 2.3.2. Adipokine Concentration

Two-way RM ANOVA demonstrated that there was no significant effect of the BODY COMPOSITION factor on the concentration of visfatin, leptin, resistin, adiponectin, asprosin, or irisin (*p* > 0.05). There was a significant effect of the EXERCISE factor on the concentration of visfatin (*p* < 0.01), resistin (*p* < 0.01), and asprosin (*p* = 0.02). There was a significant interaction of the BODY COMPOSITION × EXERCISE factors on the change in visfatin concentration (*p* = 0.04) (Table 3).

Post hoc analysis (Tukey’s HSD test) indicated a significant increase in resistin concentration after maximal intensity exercise in all groups (NFAT-NLBM, *p* < 0.01; NFAT-HLBM, *p* = 0.02; HFAT-NLBM, *p* < 0.01). Additionally, a significant increase in visfatin concentration was observed only in the NFAT-NLBM (*p* < 0.01) group (Figure 2, Appendix A).

### 2.4. Correlations

A significant positive low correlation was found between the post-exercise change in AOPP concentration and LBM (Spearman’s test, r = 0.33, *p* < 0.05), and also between the post-exercise change in AOPP concentration and BM (Pearson’s correlation coefficient, r = 0.31, *p* < 0.05). A negative low correlation was observed between the post-exercise change in irisin concentration and BM (Pearson’s test, r = −0.35, *p* < 0.05), and a positive moderate correlation was found between the change in visfatin concentration after exercise and VO2max × BM−1 (Spearman’s test, r = 0.50, *p* < 0.05).

## 3. Discussion

In our study, the effect was demonstrated of maximal effort on six of the twelve parameters investigated (LPO, CAT, SOD, visfatin, resistin, and asprosin), while the effect of body composition on all investigated parameters, except leptin (%dPV uncorrected), the concentration of which was higher in people with increased fat tissue content, was insignificant.

We found that maximal intensity aerobic exercise induces OS associated with lipid oxidation and affects the induction of resistin secretion in young healthy men, regardless of their body composition, while the concentration of visfatin increased only in the group with average body composition.

Individuals with increased body fat responded to maximal intensity exercise, also with a significant decrease in SOD activity and an increase in CAT activity.

In our study, we also showed that the decrease in plasma volume under the influence of maximal exercise was significant and did not differ between groups. In the context of correct calculation for concentrations of biochemical indicators, it is necessary to consider changes resulting from the transfer of water between vascular and extravascular spaces resulting, in particular, from intense exercise.

### 3.1. Effect of Exercise on Prooxidant–Antioxidant Balance Indices

The treadmill test conducted in the present study was a graded effort performed until refusal due to exhaustion. It was a high-intensity exercise, maximally engaging aerobic and anaerobic energy processes [5]. Research regarding the effect of an intensive exercise protocol on prooxidant and antioxidant status is ambiguous [8,16]. Data obtained from meta-analyses indicate that the occurrence of OS due to a single intensive effort positively correlates with the intensity and duration of exercise, and the consequences of such a state pass within 24 h, without having further long-term effects [16,17]. It is most often noted that maximal effort causes OS with the accompanying mobilization of antioxidant defense [18]. ROS, generated as a result of myocyte contractile activity and on the basis of hormesis, may stimulate the signaling of pathways, such as nuclear factor-kappa B (NF-κb), proliferator-activated receptor-γ coactivator-1α (PGC-1α), or nuclear factor erythroid 2-related factor 2 (Nrf2), to produce antioxidant proteins or increased mitochondrial biogenesis [8,19,20].

Systemic OS may occur as a result of both running, and cycling on a cycle ergometer [5,21]. The limiting intensity of exercise causing oxidative stress, in particular, lipid oxidation, is considered to be ~70 to 75% VO_2max_ [16,22,23]. As noted by the researchers, OS markers in healthy individuals after various short- or medium-duration exercises occur within five minutes after the end of the exercise and remain elevated for about 30 min, even up to 24 h, although individual indicators reach peak values at different times; hence, the need for a broad analysis at multiple time points [3,6,16,24].

Although there are no similar studies in which different body composition among healthy individuals would be taken into account, it may be assumed that people who train, even those who are amateurs, are characterized by a higher content of muscle mass than their non-training peers. Therefore, comparisons can be made between available data also considering body composition indices.

In the current study, differences were not shown in the values of prooxidant–antioxidant balance indices between groups differing in body composition. It should be noted that participants in all groups, despite differences in body composition, were characterized by a normal BMI range and a comparable level of physical performance defined by descriptive categories. The subjects were allocated to the category of good and high physical performance. As previously demonstrated, OSI does not correlate with the level of VO_2_max; however, it does correlate negatively with the level of physical activity [25].

In the present research, the effect has been demonstrated of a maximal effort on the change in oxidative reaction parameters (LPO) and antioxidant enzymes (CAT, SOD). Depletion of antioxidants and an increase in oxidants leads, especially, to lipid peroxidation processes in muscle cells [26]. In our study, in each of the groups, there was an increase in LPO, which indicates the susceptibility of lipids to damage under the influence of a maximal effort. This is consistent with previous data [27,28] and, at the same time, partially contradictory to other data indicating no reported changes in OS markers after exercise in recreationally active individuals [29,30]. Finkler et al. [28] found a slow increase in catalase activity during the resting period, as well as an immediate increase in lipid peroxidation markers—reaction products with thiobarbituric acid (TBARS). Ajmani et al. [31] noted, in addition to the increase in the TBARS, a rise in erythrocyte membrane stiffness after acute exercise on a treadmill due to the peroxidation of membrane lipids. Other researchers [32] observed that under the influence of a marathon, the concentration of LPO increases while the activity of SOD decreases, which is consistent with the results obtained in the present study. Similarly, Jammes et al. [33] demonstrated a transient short-term increase in lipid peroxidation after graded exercise in healthy men, which was correlated with VO_2_max.

In scientific research, an increase in protein oxidation was observed in a group of young people under the influence of anaerobic exercise [34]. Also, a single bout of high-intensity and graded physicals exercise performed until refusal due to exhaustion caused an increase in protein and lipid oxidation [35,36]. The authors of the presented work demonstrated a positive correlation between total and lean body mass and post-exercise changes in the marker of oxidative protein damage, as well as a significant increase in AOPP concentration after exhaustive exercise in the group characterised by in-creased muscle mass. It should be noted, however, that this only concerned the situation when the reduction in plasma volume induced by exercise was not taken into account.

Analysis of our results showed the effect of exercise on changes to plasma antioxidant parameters in the high adipose tissue group, but their directionality is ambiguous. While SOD activity decreased, an increase in catalase activity was detected. As suggested by some authors, the increase in CAT due to exercise may be a protective mechanism of the tissues against hydrogen peroxide generation [37]. In our control group and that with increased muscle mass, no changes in enzyme activity were noted, which is consistent with selected studies [6,21,38], but inconsistent with others [27]. In the research carried out by Djordevic et al., increases in the concentrations of superoxide anion, hydrogen peroxide, and nitric oxide were observed in untrained individuals [38]. This occurred under the influence of exercise without disturbing the antioxidant status. Importantly, large amounts of the superoxide anion were produced as a result of exercise react with nitric oxide to form peroxynitrite, leading to the inactivation of enzymes and modification of cysteine residue thiol groups [8,19]. Perhaps such a mechanism occurred in our study. In contrast to previous studies [34], the authors of the present study cannot confirm the effect of graded maximal exercise on DNA nucleic acid damage. This fact may be related to the measurement of ox/DNA/RNA products in plasma and not at the level of isolated cells, or in urine. It is also possible that the exercise duration was too short to induce oxidative damage.

### 3.2. Influence of Exercise on Adipokine Levels

The size of adipose tissue hormone secretion as a result of physical activity is deter-mined by the content of fat mass, the type of exercise, its duration, and the balance of energy expenditure. For example, in the case of leptin, the amount of energy expenditure seems to be of the greatest significance [39]; hence, in efforts lasting a few to a dozen or so minutes, no changes in its concentration in the blood are usually noted [40,41,42,43]. This is also confirmed by the results of the current study. In longer efforts, a decrease in both leptin level [44] and mRNA expression of the *ob* gene was observed. The decrease immediately after exercise and at subsequent time points following the effort, despite the lack of weight loss, may be related to the increase in the production of non-esterified fatty acids (NEFA) that negatively correlate with leptin levels [45]. However, in some studies, the possibility has been indicated of a postexercise delay in leptin secretion among healthy individuals with increased adipose tissue [43]. In the context of leptin function and stimulation of the satiety center in the hypothalamus, this may therefore have a positive impact on the regulation of food intake [46].

Another adipokine that significantly regulates carbohydrate metabolism, including insulin sensitivity and lipid metabolism, is adiponectin [47]. Covering a distance of 6000 m on rowing ergometer with an average heart rate of 90%HRmax (~20 min) induced a decrease in adiponectin concentration by 8.1% (not significant). Nonetheless, taking changes in plasma volume into account, the decrease was significant (−11.3%). However, no changes in leptin concentration were observed. Interestingly, 30 min after the completion of exercise, the adiponectin level increased by 19.3% and 20% (without and with plasma volume correction, respectively) [48]. It is worth noting that the tested athletes had an adipose tissue of 10.3%. However, in other studies, no changes in this adipokine or leptin were demonstrated after 60 min of exercise in healthy men, which may have been due to the relatively low load used, which was 65% of VO_2_max [41]. Similarly, in the trial by Bobbert et al. [47], no significant change in this adipokine was observed after a test carried out at a gradually increasing intensity on a cycloergometer among men with a normal BMI. However, the authors did not directly assess changes in plasma volume, but as they emphasized, no change in its osmolality occurred, which could mean a lack of significant changes in PV [47]. In the present study, a significant increase in adiponectin concentration was noted for all groups, but excluding changes in PV, but when correction was applied, the changes turned out to be insignificant. The lack of exercise effect on the level of this adipokine may also have been due to the inhibitory effect of catecholamines released during maximal exercise [49]. Raising the level of adiponectin is therefore possible mainly through significant changes in body composition [50].

In the current research, a visfatin concentration increase was noted in the group with average adipose and average muscle tissues content, both with and without correction for PV changes. In a trial conducted on rowers, a decrease in visfatin concentration was noted after two hours of moderate intensity exercise [51], while sprinting exercise among young healthy people caused an almost two-fold increase in the hormone, and this was after taking changes in plasma volume into account. Visfatin mRNA expression increases in adipocytes of abdominal adipose tissue, without a concomitant increase in blood concentration after aerobic exercise, which may indicate paracrine regulation of this hormone [52]. Although its biological functions are not fully understood, its action results in increased insulin sensitivity of peripheral tissues (through insulinomimetic properties), while its paracrine action may promote the increase in obesity through differentiation of fat cells and intensification of adipogenesis and up-regulation of other pro-inflammatory cytokines, such as IL-1β, IL-6, or TNF-α in vascular endothelial cells and peripheral blood monocytes [53,54]. It is very interesting that the rise in visfatin concentration may promote the transition of the body into an anorexigenic state, as a result of which appetite is inhibited via the POMC/CART melanocortin pathway [55].

The authors of the current research also noted a significant effect of the maximal effort on resistin concentration. In all groups, a comparable increase in resistin concentration was obtained. Huminska-Lisowska observed a 55% increase in resistin concentration, but only in the group with an adipose tissue level above 14% [43]. Resistin, by promoting an increase in tissue resistance to insulin, inhibits the decrease in blood glucose levels due to vigorous exercise. Contrary results were achieved in the study by Fortes et al., 2024 [56]. These authors noted a decrease in resistin after a single strength training session in obese men, as well as some data by confirming a decrease following a 45 min exercise intervention (85% VO_2_max) in healthy young men [56,57]. In contrast, no changes were noted following high-intensity aerobic exercise among obese men [50]. The post-exercise increase in resistin concentration is most often achieved among people with higher physical performance. It is also related to the amount of energy expenditure and, to a lesser extent, to the content of adipose tissue itself [58,59]. Although our studies do not confirm this, we did not find any correlation between the post-exercise change in resistin concentration and VO_2_max.

The authors of the present study observed a decrease in asprosin concentration after maximal exercise when the results were analyzed with the PV correction taken into account, but in total for all groups, which is consistent with the research results obtained by Ceylan et al. [60], who observed a similar effect after 30 min of low-intensity aerobic exercise, but in people with both normal BMI and obese people. But, after analyzing the changes in asprosin concentration in each group separately, no significant effect of maximal exercise on the level of this hormone in the blood was found, regardless of the body composition of the subjects studied. In another study, a decrease in asprosin was noted in both in the case of aerobic and high-intensity interval exercise, but its level was greater in obese people [61]. The authors emphasize that higher exercise intensity leads to a greater decrease in the level of this adipokine [61]. However, it was also shown that the concentration of asprosin in the blood does not change after a single exercise bout [62]. Moreover, an increase in asprosin was noted under the influence of short anaerobic exercise in the study by Wiecek et al. [63]. Physiologically, asprosin is responsible for appetite stimulation and glucose release from the liver, and elevated values are noted among obese people [64].

The effect of exercise on the concentration of irisin was not seen in the current study. This myokine regulating the browning of white adipose tissue and energy expenditure comes from muscle tissue in about 62%, the rest is secreted by adipose tissue. Its role in exercise is not fully understood [65]. Despite individual meta-analyses [66] in which an average 15% increase was demonstrated in irisin concentration after a single exercise bout, there is no convincing evidence for a post-exercise increase in irisin concentration. Moreover, in studies for which an increase was exhibited in concentration, changes in plasma volume were not taken into account [67,68]. There is also a possible reason for the differentiation of irisin secretion depending on the type of exercise, as vigorous strength training caused an increase in irisin precursors, i.e., *FNDC5* gene mRNA in the muscles of young men, in contrast to aerobic exercise or the entire training process [69].

### 3.3. Limitations of the Study

In our work, we examined peripheral effects of physical exercise in groups with different body compositions; however, future directions of research should also concern the secretory role of adipocytes and myocytes at the cellular level.

A greater number of post-exercise measurements at different time points should also be taken into account, as well as the amount of energy expenditure caused by such an effort. It is also worth considering the gender and age of the participants.

It has been shown that with similar oxidative damage to proteins and lipids in women and men after maximal intensity exercise, only men experience an increase in the oxidative stress index, which results from their higher VO_2_max and greater intensity of anaerobic changes during graded exercise. In women, after maximal intensity exercise, the oxidative stress index does not rise due to the increased antioxidant capacity of the plasma [2]. It was found that supramaximal exercise induced an increase in asprosin and irisin secretion while reducing leptin secretion, but only in women. The post-exercise increase in irisin concentration was positively correlated with the percentage fat content, while being negatively correlated with total and lean body mass [63].

The reaction may also depend on the age of the subjects, due to the reduction in the antioxidant capacity of the blood in older people. The expression of antioxidant enzymes is down-regulated; i.e., their concentration as well as activity decreases. The total antioxidant capacity of ROS sweeping capacity decreases while total oxidative status is increased [70].

Other inflammatory cytokines secreted by adipose tissue, especially during acute physical exercise, were also not the subject of this research.

## 4. Materials and Methods

The study included participant qualification, dietary recommendations, gradually increasing exercise load until refusal (maximal intensity exercise), biochemical determinations of adipokines, and markers of the prooxidant–antioxidant balance.

The results obtained for young, healthy, non-smoking, physically active but untrained men with different body composition were analyzed, selected after analysis of somatic measurement results from the general population of 1549 men aged 18–30.

The study was conducted in accordance with the 1964 Declaration of Helsinki, after obtaining the consent of the Bioethics Committee at the Regional Medical Chamber (88/KBL/OIL/2010). Participants were informed in writing about the purpose and plan of the study and about possible inconveniences and risks associated with the protocol, and then provided their written informed consent to participate in the study. Participants could withdraw from the study at any stage of its implementation without giving reason. Before beginning the study, the participants were familiarized with the laboratory conditions.

### 4.1. Participant Qualification

Somatic measurements were performed in a group of 1549 men, as previously undertaken [4] using the method of bioelectrical impedance (BIA) with an eight-electrode, multi-frequency (5, 50, 200 kHz) analyzer (Ja-won IOI-353 Body Composition Analyzer, Gyeongsa, Korea), and body mass (BM), total fat mass (FAT), lean body mass (LBM), body fat percentage (%FAT) were measured. Body composition was assessed at a normal body hydration level (euhydration), at a similar external temperature (22–24 °C) and in a standing position, at an angle of about 30° between the upper limbs extended at the elbow joints and the trunk. The subjects were dressed only in briefs; a body mass correction of 300 g was assumed. Analyzer electrodes were placed in the hands (two each) and under the feet (two each), which were degreased and dried. Body height (BH) was measured using the Martin anthropometer to the nearest 1 mm (Vitako, Szczecin, Poland). Body mass index (BMI) was also calculated.

The individuals were qualified into one of three groups:NFAT-NLBM group—men with average %FAT and average LBM;NFAT-HLBM group—men with average %FAT and high LBM;HFAT-NLBM group—men with high %FAT and average LBM.

Average values were considered to be between the 40th and 60th percentile, which were 14.0–18.5% for %FAT and 59.0–64.3 kg for LBM. High values were considered to be above the 80th percentile, which were >21.5% for %FAT and >66.3 kg for LBM.

Men with chronic diseases, professionally training sports, smoking, applying a specific diet and/or supplements, or those with contraindications to perform maxima intensity physical activity were excluded from the study.

Men who met the criteria for inclusion in one of the groups and were willing to participate in the exercise testing underwent medical qualification (medical interview, blood pressure measurement, stress ECG, blood count, fasting glucose, lipid profile, and glycated hemoglobin panels).

The study included 49 volunteers, of whom seven withdrew from the study. Ultimately, the results obtained for 42 males were analyzed.

The characteristics of the subjects are presented in Table 1.

### 4.2. Diet Control

For seven days prior to the IT, the participants followed a diet (approximately 2700 kcal/day) created by a dietician, standardized in terms of the percentage of proteins (15%), fat (30%), and carbohydrates (55%) to cover the energy requirement and in terms of the content of antioxidant vitamins: A (630 µg/day), E (10 mg/day), and C (75 mg/day). The diet was verified based on the analysis of seven-day food diaries kept by the participants using the “Album of photographs of products and dishes” [71].

### 4.3. Exercise Protocol

A running test with a gradually increasing load was performed on a mechanical treadmill at an inclination angle of 0° (Saturn/h/p/Cosmos, Nussdorf-Traunstein, Germany) according to the scheme: four minutes at a speed of 7 km·h^−1^, then every two minutes, the running speed was increased by 1.2 km·h^−1^, until the participant was unable to continue the effort or when, despite the increase in running speed, no increase in oxygen uptake per minute (VO_2_) was recorded. The stress tests were performed in the morning hours (9:00–11:00 a.m.), at a similar ambient temperature (20–22 °C), following at least eight hours of overnight rest, two hours after consuming a light meal, and in a state of proper hydration. The participants did not perform any intense physical efforts, they did not consume alcohol or products containing caffeine or other stimulants 24 h prior to the stress test or on the day of its execution. The stress tests were performed under medical supervision.

### 4.4. Blood Sampling and Biochemical Analyses

Five minutes before (T0) and three minutes after completing the exercise (T1), venous blood was collected from the elbow area using a BD Vacutainer^®^ vacuum system (Becton Dickinson, Franklin Lakes, NJ, USA). The following concentrations were determined in the blood plasma: advanced protein oxidation products (AOPP), markers of oxidative damage to DNA/RNA (DNA/RNA/ox), markers of lipid peroxidation (LPO)—K2EDTA tubes; catalase (CAT), superoxide dismutase (SOD), and glutathione peroxidase (GPx) activity—lithium heparin tubes, as well as leptin, resistin, adiponectin, asprosin, and irisin—K2EDTA tubes and a protease inhibitor: aprotinin 0.6 TIU/1 mL of blood. The concentration of visfatin was determined in the blood serum—tubes with a coagulation activator.

Blood was centrifuged to plasma immediately after collection, but to serum, after 20 min in order for it to clot (RCF 1.000× *g* for 15 min at 4 °C; MPW-351R, MPW Med. Instruments, Warsaw, Poland). The collected plasma and serum were stored at −70 °C (ULF 390 Arctiko low temperature freezer, Esbjerg, Denmark) until analysis.

The assays were performed in accordance with the methodology presented by the manufacturer of the reagents: Nori^®^Human Asprosin ELISA Kit GR 111426 (Genorise, Glen Mills, PA, USA). The detection range was 1.5–100 ng/mL, intra-assay CV < 6%, inter-assay CV < 9%; Human Irisin ELISA Kit RAG018R, detection range 0.001–5 μg/mL, intra-assay CV = 6.9%, inter-assay CV = 9.1%; Human Leptin ELISA Kit RD191001100, detection range of 1–50 ng/mL, intra-assay CV = 5.9%, inter-assay CV < 5.6%; Human Adiponectin ELISA Kit RD191023100, detection range 1–150 ng/mL, intra-assay CV = 3.9%, inter-assay CV = 6.0%; Human Resistin ELISA Kit RD191016100, detection range between 1 and 50 ng/mL, intra-assay CV = 5.9%, inter-assay CV = 7.6%; Human Visfatin (NAMPT) ELISA Kit RAG004R, detection range 0.125–8.0 ng/mL, intra-assay CV < 9.1%, inter-assay CV < 7.2% (BioVendor, Karasek, Czech Republic); AOPP photometric test KR7811W, detection range of 6.25–100 μmol/L, intra-assay CV < 5.6%, inter-assay CV < 16.6%; PerOx KC5100 colorimetric test for LPO, detection range 7–800 μmol/L, intra-assay CV < 2.9%, inter-assay CV < 6.6% (Immundiagnostik AG, Bensheim, Germany); DNA/RNA oxidative damage 589320 ELISA kit, detection range between 10.3 and 3000 pg/mL, intra-assay CV < 11.6%, inter-assay CV < 10.7%; and antioxidant enzyme activity was determined using the SOD 706002, GPx 703102 and CAT 707002 kits (Cayman Chemical Company, Ann Arbor, MI, USA). The detection range was 0.025–0.25 U/mL for SOD, 0.05–0.344 nmol/min/mL for GPx and 2–34 nmol/min/mL for CAT.

In the whole blood, hemoglobin (HGB) concentration was determined spectrophotometrically by the cyanide-methemoglobin method using Drabkin’s reagent (Pol-Aura, Morąg, Poland). Hematocrit (HCT) level was established via the microhematocrit method in triplicate and the mean results were calculated.

Percentage changes in plasma volume (%ΔPV) were calculated from HGB concentration and HCT values according to the Dill and Costilla equation, modified by Harrison et al. [72,73]. The post-exercise concentrations of the analyzed adipokines and prooxidant–antioxidant balance markers were corrected (T1_PV_) according to the formula proposed by Kraemer and Brown [74].

### 4.5. Statistical Analysis

The statistical significance of differences was assumed for the level of *p* < 0.05.

Distribution of the results for the analyzed variables was checked using the Shapiro–Wilk test, and the equality of variance with Levene’s test. For single measurements (age, body composition, physical fitness, blood count, glucose, lipid profile, and inflammatory markers), the significance of group-related differences was assessed via independent-sample tests, applying ANOVA, one-way analysis of variance or the Kruskal–Wallis test (Table 1 and Table 2), followed by Tukey’s HSD test (for BMI, FAT, LBM, and VO_2_max × BM^−1^) or Dunn’s test (for BM) [75].

The sample size was calculated for three groups, two measurements, and the power of the test was 1− β = 0.80, *p* = 0.05 and medium effect size was f = 0.25 (η^2^ = 0.06) [76]. For the assumptions adopted in this way, a total sample size = 42 was obtained (G*Power 3.1.9.6, Franz Faul, Universitat Kiel, Germany).

Two-way analysis of variance with repeated measures (two-way RM ANOVA) was used to examine the influence of the main factors, i.e., BODY COMPOSITION, EXERCISE, and the interaction of BODY COMPOSITION × EXERCISE, on the level of the analyzed biochemical markers in blood: (1) without correction for %dPV of post-exercise values (T0 and T1 in each of the three groups: NFAT-NLBM, NFAT-HLBM, HFAT-NLBM), as well as additionally (2) with correction for %dPV of post-exercise values (T0 and T1_PV_ in each of the three groups: NFAT-NLBM, NFAT-HLBM, HFAT-NLBM). Effect sizes for ANOVA were calculated using partial eta squared (η^2^) and interpreted as 0.010–0.059 = small, 0.060–0.139 = medium, >0.14 = large (Table 3).

When significant influence of the main factors was found, post hoc analysis was performed using Tuckey’s HSD test [75] (Figure 1 and Figure 2, Appendix A).

For changes in the level of specific variables after exercise, confidence intervals were determined (95% CI) (Appendix A).

Pearson’s or Spearman’s correlation coefficients were calculated between BM, BMI, LBM, %FAT, and VO_2_max and post-exercise changes in the concentration of AOPP, DNA/RNA/ox, LPO, visfatin, leptin, resistin, adiponectin, asprosin, irisin as well as SOD, CAT, and GPx activity (%dPV corrected). The following correlation assessment was adopted depending on the value of the r correlation coefficient: no correlation if r ≤ 0.19, low correlation if 0.2 ≤ r ≤ 0.39, moderate correlation if 0.40 ≤ r ≤ 0.59, moderately high correlation if 0.6 ≤ r ≤ 0.79, and high correlation if r ≥ 0.8

The STATISTICA 13.3 package (StatSoft, Inc., Tulsa, OK, USA) was used for calculations.

## 5. Conclusions

In our study, the effect of maximal effort on six (LPO, CAT, SOD, visfatin, resistin, and asprosin) of the twelve parameters investigated was demonstrated, while the effect of body composition on all parameters investigated (except %dPV uncorrected leptin) was insignificant. Maximal intensity aerobic exercise induces significant increases in resistin secretion in people with above-average body fat, in people with above-average muscle mass and in people with average body composition. Maximal physical exercise damages lipids regardless of the exercising subjects’ body composition. Large fat tissue content predisposes to exercise-induced disorders in the activity of antioxidant enzymes.

We have also shown that it is necessary to take changes in blood plasma volume into account in the assessment of post-exercise biochemical marker levels.

## Figures and Tables

**Figure 1 ijms-26-00350-f001:**
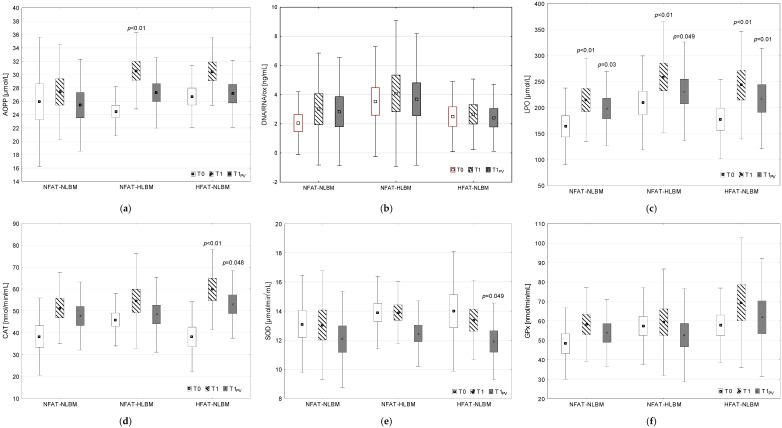
Concentration of prooxidant–antioxidant balance markers before (T0), as well as after maximal intensity exercise, without correction (T1) and after correction for percentage plasma volume changes (T1_PV_): (**a**) AOPP—advanced protein oxidation products; (**b**) DNA/RNA/ox—markers of oxidative damage to nucleic acids; (**c**) LPO—markers of lipid peroxidation; (**d**) CAT—catalase; (**e**) SOD—superoxide dismutase; (**f**) GPx—glutathione peroxidase; *p* < 0.05—statistically significant difference T1 vs. T0 and T1_PV_ vs. T0 in each group (post hoc Tukey’s HSD test following two-way repeated measures analysis of variance for data (1) without correction for %dPV of post-exercise values and independently (2) with correction for %dPV of post-exercise values); NFAT-NLBM—group with average content of %FAT and with average content of LBM, NFAT-HLBM—group with average content of %FAT and above-average content of LBM, HFAT-NLBM—group with above-average content of %FAT and average content of LBM; marker—mean, box—standard error (SE), whisker—standard deviation (SD).

**Figure 2 ijms-26-00350-f002:**
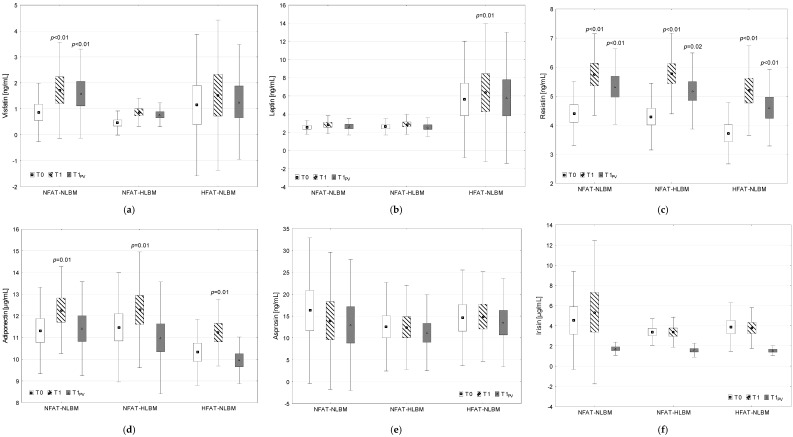
Concentration of adipokines: visfatin (**a**), leptin (**b**), resistin (**c**), adiponectin (**d**), asprosin (**e**), and irisin (**f**), before (T0), as well as after maximal intensity exercise without (T1) and after correction for percentage plasma volume changes (T1_PV_); *p* < 0.05—statistically significant difference T1 vs. T0 and T1_PV_ vs. T0 in each group (post hoc Tukey’s HSD test following two-way repeated measures analysis of variance for data (1) without correction for %dPV of post-exercise values and independently, (2) with correction for %dPV of post-exercise values); NFAT-NLBM—group with an average content of %FAT and with an average content of LBM, NFAT-HLBM—group with an average content of %FAT and above-average content of LBM, HFAT-NLBM—group with above-average content of %FAT and average content of LBM; marker—mean, box—standard error (SE), whisker—standard deviation (SD).

**Table 1 ijms-26-00350-t001:** Age, body composition, and physical performance of participants.

	GROUP			One-Way ANOVA	Kruskal–Wallis Test
VARIABLE	NFAT-NLBM Mean ± SD (N = 13)	NFAT-HLBM Mean ± SD (N = 16)	HFAT-NLBM Mean ± SD (N = 13)	*p*-Value (F)	*p*-Value (H)
Age (years)	21.23 ± 1.42	21.50 ± 2.10	21.62 ± 2.90		0.89 (0.23)
BM (kg)	73.35 ± 2.34	85.06 ± 5.28 *	80.19 ± 4.40 *		<0.01 (25.46)
BMI (kg × m^−2^)	22.94 ± 1.33	24.76 ± 1.54 *	25.36 ± 1.12 *	<0.01 (11.35)	
%FAT (%)	16.27 ± 1.60	16.86 ± 2.45	23.09 ± 1.89 *^#^	<0.01 (45.40)	
LBM (kg)	61.38 ± 1.29	70.74 ± 4.97 *	61.62 ± 2.68 ^#^	<0.01 (34.56)	
VO_2_max × BM^−1^ (mL × kg^−1^ × min^−1^)	58.23 ± 5.84	52.94 ± 5.13 *	50.25 ± 4.57 *	<0.01 (7.97)	

BM—body mass; BMI—body mass index; %FAT—percentage of body fat; LBM—lean body mass; VO_2_max—maximal oxygen uptake per minute; NFAT-NLBM—group with average content of %FAT and with average content of LBM, NFAT-HLBM—group with average content of %FAT and above-average content of LBM, HFAT-NLBM—group with above-average content of %FAT and average content of LBM; SD—standard deviation; post hoc analysis: Tukey’s HSD test (for BMI, FAT, LBM, and VO_2_max × BM^−1^) or Dunn’s test (for BM): * statistically significant difference (*p* < 0.05) compared to NFAT-NLBM group, ^#^ statistically significant difference (*p* < 0.05) compared to NFAT-HLBM group.

**Table 2 ijms-26-00350-t002:** Medical qualification of participants: blood count, glucose, lipid profile, and inflammatory markers (ESR, CRP).

	GROUP			One-Way ANOVA	Kruskal–Wallis Test
VARIABLE	NFAT-NLBM Mean ± SD (N = 13)	NFAT-HLBM Mean ± SD (N = 16)	HFAT-NLBM Mean ± SD (N = 13)	*p*-Value (F)	*p*-Value (H)
Erythrocytes (10^6^ × μL^−1^)	4.97 ± 0.34	5.06 ± 0.23	5.16 ± 0.22	0.19 (1.72)	
Hemoglobin (g × dL^−1^)	14.97 ± 1.17	15.12 ± 0.74	15.50 ± 0.66	0.29 (1.28)	
Hematocrit (%)	43.77 ± 2.79	44.31 ± 1.65	45.34 ± 1.88	0.17 (1.83)	
Leukocytes (10^3^ × μL^−1^)	5.43 ± 0.95	5.87 ± 0.98	5.74 ± 1.26	0.53 (0.64)	
Neutrophils (%)	56.62 ± 11.90	53.79 ± 6.95	53.38 ± 6.89	0.59 (0.54)	
Lymphocytes (%)	35.12 ± 8.46	36.28 ± 7.94	36.13 ± 6.63	0.91 (0.09)	
Monocytes (%)	7.42 ± 1.49	7.19 ± 2.23	7.77 ± 1.51	0.70 (0.36)	
Eosinophils (%)	2.88 ± 1.98	2.33 ± 1.32	2.37 ± 1.58		0.79 (0.47)
Basophils (%)	0.23 ± 0.10	0.28 ± 0.20	0.22 ± 0.10		0.51 (1.35)
Platelets (10^3^ × μL^−1^)	216.0 ± 46.8	236.9 ± 48.3	232.6 ± 34.5	0.43 (0.87)	
Glucose (mmol × L^−1^)	4.10 ± 0.30	4.12 ± 0.45	4.40 ± 0.38		0.13 (4.02)
Total cholesterol (mmol × L^−1^)	4.21 ± 0.67	4.28 ± 0.56	4.47 ± 0.95	0.65 (0.43)	
HDL-cholesterol (mmol × L^−1^)	1.37 ± 0.20	1.34 ± 0.18	1.40 ± 0.30	0.77 (0.26)	
LDL-cholesterol (mmol × L^−1^)	2.43 ± 0.73	2.54 ± 0.49	2.61 ± 0.82	0.79 (0.24)	
Triglycerides (mmol × L^−1^)	0.73 ± 0.27	0.90 ± 0.26	1.00 ± 0.50		0.20 (3.25)
Glycated hemoglobin (%)	5.14 ± 0.33	5.12 ± 0.74	5.12 ± 0.27		0.95 (0.09)
ESR (mm × h^−1^)	2.08 ± 0.28	3.31 ± 1.58	3.23 ± 2.20		0.03 (7.07)
CRP (mg × L^−1^)	0.36 ± 0.20	0.84 ± 0.80	0.79 ± 0.62		0.16 (3.64)

HDL—high-density lipoprotein; LDL—low-density lipoprotein; ESR—erythrocyte sedimentation rate; CRP—C-reactive protein; NFAT-NLBM—group with an average content of FAT and with an average content of LBM, NFAT-HLBM—group with an average content of FAT and above-average content of LBM, HFAT-NLBM—group with above-average content of FAT and average content of LBM; SD—standard deviation; post hoc analysis: (Dunn’s test for ESR) showed that the ESR was comparable in all groups (*p* > 0.05).

**Table 3 ijms-26-00350-t003:** Assessment of body composition and maximal exercise influence on blood prooxidant–antioxidant balance parameters and adipokine concentration in young men—results of two-way repeated measures analysis of variance (two-way RM ANOVA).

		Results of Two-Way RM ANOVA		
		BODY COMPOSITION	EXERCISE	BODY COMPOSITION × EXERCISE
VARIABLE	Measuring Points	F (*p*) η^2^	F (*p*) η^2^	F (*p*) η^2^
AOPP	(1) %dPV uncorrected	0.37 (0.69) 0.02	26.79 (<0.01) 0.41	3.44 (0.04) 0.15
(μmol × L^−1^)	(2) %dPV corrected	0.18 (0.83) 0.01	1.73 (0.20) 0.04	2.17 (0.13) 0.10
DNA/RNA/ox	(1) %dPV uncorrected	0.69 (0.51) 0.03	3.31 (0.08) 0.08	0.53 (0.59) 0.03
(ng × mL^−1^)	(2) %dPV corrected	0.64 (0.53) 0.03	1.01 (0.32) 0.03	0.80 (0.46) 0.04
LPO	(1) %dPV uncorrected	093 (0.40) 0.05	76.60 (<0.01) 0.66	0.62 (0.54) 0.03
(μmol × L^−1^)	(2) %dPV corrected	0.81 (0.45) 0.04	28.72 (<0.01) 0.42	0.81 (0.45) 0.04
CAT	(1) %dPV uncorrected	0.56 (0.58) 0.03	23.38 (<0.01) 0.37	1.69 (0.20) 0.08
(nmol × min^−1^ × mL^−1^)	(2) %dPV corrected	0.38 (0.69) 0.02	10.28 (<0.01) 0.21	1.82 (0.18) 0.09
SOD	(1) %dPV uncorrected	0.31 (0.74) 0.02	0.38 (0.54) 0.01	0.26 (0.77) 0.01
(μmol × min^−1^ × mL^−1^)	(2) %dPV corrected	0.16 (0.85) 0.01	15.46 (<0.01) 0.28	0.57 (0.57) 0.03
GPx	(1) %dPV uncorrected	0.87 (0.43) 0.04	3.99 (0.05) 0.09	0.63 (0.54) 0.03
(nmol × min^−1^ × mL^−1^)	(2) %dPV corrected	0.72 (0.49) 0.04	0.21 (0.65) 0.01	0.83 (0.44) 0.04
Visfatin	(1) %dPV uncorrected	0.67 (0.52) 0.03	31.32 (<0.01) 0.45	2.33 (0.11) 0.11
(ng × mL^−1^)	(2) %dPV corrected	0.71 (0.50) 0.04	17.15 (<0.01) 0.31	3.36 (0.04) 0.15
Leptin	(1) %dPV uncorrected	3.06 (0.05) 1.14	14.36 (<0.01) 0.27	1.74 (0.19) 0.08
(ng × mL^−1^)	(2) %dPV corrected	2.94 (0.06) 0.13	0.73 (0.40) 0.02	0.48 (0.62) 0.02
Resistin	(1) %dPV uncorrected	1.03 (0.37) 0.05	226.07 (<0.01) 0.85	0.24 (0.79) 0.01
(ng × mL^−1^)	(2) %dPV corrected	1.34 (0.27) 0.16	102.11 (<0.01) 0.72	0.02 (0.98) <0.01
Adiponectin	(1) %dPV uncorrected	1.78 (0.32) 0.06	38.70 (<0.01) 0.50	0.09 (0.91) <0.01
(µg × mL^−1^)	(2) %dPV corrected	1.51 (0.23) 0.07	1.98 (0.17) 0.05	0.90 (0.42) 0.04
Asprosin	(1) %dPV uncorrected	0.20 (0.82) 0.01	0.89 (0.35) 0.02	1.09 (0.35) 0.05
(ng × mL^−1^)	(2) %dPV corrected	0.22 (0.80) 0.01	6.34 (0.02) 0.14	0.75 (0.48) 0.04
Irisin	(1) %dPV uncorrected	0.09 (0.91) <0.01	11.15 (<0.01) 0.22	0.48 (0.62) 0.02
(µg × mL^−1^)	(2) %dPV corrected	0.13 (0.88) 0.01	1.44 (0.24) 0.04	0.86 (0.43) 0.04

AOPP—advanced protein oxidation products; DNA/RNA/ox—markers of oxidative damage to nucleic acids DNA and RNA; LPO—markers of lipid peroxidation; CAT—catalase; SOD—superoxide dismutase; GPx—glutathione peroxidase; %dPV—percentage changes in plasma volume; two-way RM ANOVA—two-way repeated measures analysis of variance: (1) includes results without correction for %dPV of post-exercise values (results of 6 groups: NFAT-NLBM T0, NFAT-NLBM T1, NFAT-HLBM T0, NFAT-HLBM T1, HFAT-NLBM T0, HFAT-NLBM T1); (2) includes results with correction for %dPV of post-exercise values (results of 6 groups: NFAT-NLBM T0, NFAT-NLBM T1_PV_, NFAT-HLBM T0, NFAT-HLBM T1_PV_, HFAT-NLBM T0, HFAT-NLBM T1_PV_).

## Data Availability

The datasets used and/or analyzed during the current study are available from the corresponding author upon reasonable request.

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
