# Peer review of "Maximal Intensity Exercise Induces Adipokine Secretion and Disrupts Prooxidant–Antioxidant Balance in Young Men with Different Body Composition"

_ijms, 2025, doi:10.3390/ijms26010350_

Round 1

Reviewer 1 Report

Comments and Suggestions for Authors

The manuscript is interesting, well written and has merit, however, I have a few comments.

1.     In introduction, please do not claim that you are first, since probably most of non-English literature was not searched for.

2.     In the beginning of the discussion section, please add a paragraph summarizing main results.

3.     In the result section, the tables are very big, and it is hard to appreciate it. The authors could try to visually represent at least some of the data in figures.

4.     In sample size calculation, why did you choose large effect size? Based on what literature did you decide for large effect size, not medium or small? Since the main aim of the study was to prove there is no change between groups, this is the most important when planning the research.

5.     You only studied young men, do you think in women or older people the result would be the same?

6.     The introduction and discussion sections are in my opinion too long and not focused enough, a more concise text would be easier to follow.

7.     There are tables 1, 3, 5, where are tables 2 and 4?

8.     In tables 3 and 5, did you compare each time point independently, or did you compare the change between time points?

Reviewer 2 Report

Comments and Suggestions for Authors

In this study, the authors investigated the effects of maximal intensity exercise on certain markers of oxidative damage, antioxidant enzymes and adipokines in blood plasma (or serum in the case of visfatin) of young healthy individuals with different body compositions. The participants were divided into three groups: with average body composition, with above-average muscle mass and with above-average fat mass. In the introduction, the authors provided a comprehensive background. The objectives of the study and the research hypotheses were stated. The research design and methods were described in sufficient detail. The results were analyzed and presented in sufficient detail. Finally, the results were thoroughly discussed.

Here are my major concerns about this manuscript:

·       I start with the last sentence of the abstract (line 33), which states: “Post-exercise changes in adipokines are dependent on body composition”, while the two-way RM ANOVA showed no significant effect of the factor “Group”, i.e. body composition (Table 5), which raises the question of the accuracy of such a generalized conclusion. In addition, the factor “Group”, i.e. body composition, had no significant effect on the levels of oxidative damage markers or antioxidant enzyme activity (Table 3). For all of the following parameters: AOPP, DNA/RNA/ox, LPO, CAT, SOD, GPx, visfatin, resistin, adiponectin, asprosin and irisin (all parameters investigated except leptin), the F-values are very small and the p-values are very high. Thus, the summary of the main results of this study would therefore be that the two-way RM ANOVA showed a significant effect of the factor “Exercise” on seven of the twelve parameters investigated, while the effect of body composition on all parameters investigated (except leptin) was, to put it informally, very insignificant.

·       Lines 114-117: “The following research hypotheses have been posed: 1) maximal exercise intensity induces oxidative stress in both individuals with above-average adipose tissue and in those with above-average muscle mass; 2) maximal exercise intensity induces similar changes in adipokine secretion, regardless of body composition.” The first hypothesis implies the existence of two experimental groups, one of which has an above-average amount of adipose tissue and the other an above-average amount of muscle mass. The second hypothesis does not specify the number of experimental groups. In this manuscript, an experiment with three experimental groups was presented. The hypotheses must be formulated precisely and be consistent with the experimental design and statistical methods to allow adequate hypothesis testing.

·       The conclusions (this applies both to the Conclusions section and to the Abstract) should provide a direct answer to the hypotheses that is also consistent with the main results of the study.

·       I am interested in the choice of post-hoc tests. For the first set of results (Tables 1 and 2) one-way ANOVA or Kruskal-Wallis test, followed by Student's t-test or the Mann-Whitney U-test. For the second set of results (Table 3-6) two-way RM ANOVA, followed by Fisher's LSD test. What is the basis for the choice of post-hoc tests?

·       Lines 52-54: “In the early phase following exercise, the activity of antioxidant enzymes decreases (catalase - CAT, superoxide dismutase - SOD, glutathione peroxidase - GPx), which indicates the occurrence of oxidative stress [3].” The cited article (https://pmc.ncbi.nlm.nih.gov/articles/PMC6837471/) did not examine the activities of CAT, SOD and GPx, but rather ox-LDL, 3-NT, TAC, CK, LDH and Lac in serum. While searching through the article, I found several mentions of an increase in the activities of SOD and GPx after exercise (all in the Discussion, with reference to other articles), no mention of CAT, let alone a decrease in the activity of these three enzymes. I recommend that the authors check the appropriateness of all cited references in this manuscript.

(In addition, pagination of this article in the list of references is not correct (lines 568-569). The correct pagination is 14;22(4):176–182.)

My other remarks are:

·       It would be useful if the paper included a section on the limitations of the study.

·       Line 64: “It is known that both adipocytes and myocytes produce ROS…” All cells produce ROS.

·       Section 2.5. Correlations (lines 184-188) states that “a significant positive correlation was found”, without emphasizing the strength of the correlation. Of the four correlations mentioned, three are weak (r = 0.31-0.35) and one is moderate (r = 0.50). Statistical significance is of little importance if the correlation is weak.

·       Line 437: “Average values were considered to be between the 60th and 80th percentile…” Can the authors please explain this sentence?

·       Line 513: “For single measurements…” I suggest that the authors write clearly which results this refers to, namely age, body composition, physical fitness, blood count, glucose, lipid profile and inflammatory markers.

·       Since parametric and non-parametric tests were used to analyze the results presented in Tables 1 and 2, and Pearson and Sperm correlation coefficients were also used, I suggest that the authors indicate which tests were used for each variable (or each pair of variables in the case of correlations). This could be done either in Material and Methods or in the Results or in the Appendix.

·       Lines 521-524: I would suggest the authors write clearly that a two-way repeated measures ANOVA was used. I would also suggest renaming the factor “Group” to “Body composition”.

·       Would it be possible for the authors to create charts and put them in the Results section and move these tables (which are excellent and very informative) to the Appendix for a more descriptive and eye-catching result?

·       Several abbreviations are not introduced correctly.

·       Several typos throughout the manuscript.

·       “DNA/RNA/ox” in Table 3, while “8OHdG” in Table 4.

·       Lines 431 and 503: “Poland” is written in brackets. Shouldn't the source of the devices/materials, i.e. the manufacturer, be indicated there?

·       Line 486: Product number is RAG018R, not “RAG 018R”.

·       Lines 354-356: “In all groups, a comparable increase in resistin concentration and an increase in visfatin concentration were obtained, but only in the groups with average adipose tissue mass (NFAT-NLBM and NFAT-HLBM)”. This sentence needs linguistic revision. The meaning is not clear.

·       I would recommend a minor revision of the English language by a native English speaker.

Round 2

Reviewer 1 Report

Comments and Suggestions for Authors

The authors satisfactorily addressed my comments.

Author Response

We would like to thank the reviewer for evaluating our manuscript.

Reviewer 2 Report

Comments and Suggestions for Authors

See the attachment.

Dear authors and editors, here are my comments on the revised manuscript ijms-3315047-peer-review-v2. Please pay attention to comment 4.1.

1.       Title

As can be seen from the results (Table 3) and from the discussion, conclusions and abstract, exercise had a significant effect on 7 of the 12 parameters studied, while body composition had no significant effect on any of these parameters. Furthermore, there was virtually no significant interaction. Therefore, the title of the study should be changed to be consistent with the main results of the study.

2.       Abstract

The abstract could be further edited with regard to the narration. For example, the increase in LPO and resistin is mentioned twice, in lines 27-28 and again in lines 34-36.

3.       Introduction

The introduction of the revised manuscript has been shortened. The new introduction is well structured, concise and contains sufficient information without too many references. The aim of the study and the hypotheses are clearly formulated.

3.1.    Lines 43-44: A one-sentence paragraph is not needed. It can be merged with the next paragraph

3.2.    In lines 45-49, authors stated the following: “As a result of a single exercise bout in untrained individuals, an intensity- and time-dependent increase in the concentration of lipid peroxidation products, uric acid, antioxidant vitamins and the level of oxidative stress index (OSI), as well as a decrease in the ratio of reduced to oxidised glutathione concentration (GSH/GSSG) can be observed in the blood[2-5].”

I would suggest replacing “lipid peroxidation products” with “oxidised low density lipoprotein”. Without going into detail about lipid peroxidation and ox-LDL, only the concentration of ox-LDL was measured in the cited articles (in refs. 2 and 4).

I would also recommend replacing the word "can" with "might" or rewording the sentence. This is because these articles show that the changes mentioned are not universal, but depend on various factors (e.g. type of physical activity and gender). For example, in ref. 2, an increase in the concentration of ox-LDL is observed, whereas in ref. 4, the concentration of ox-LDL does not change in any group except women after eccentric exercise. Or the concentration of UA, which increases in ref. 3, while no changes are observed in refs. 4 and 5.

4.       Statistical analysis

4.1.    I disagree with the choice of post-hoc tests in this study.

In response to my report, the authors justified their choice of post-hoc tests with the following article:

Krzych L. Interpretation of the statistical analysis of data. Kardiochirurgia i Torakochirurgia Polska 2007; 4 (3): 315–321.

The article is also cited in the revised manuscript (lines 515 and 523).

This article states on page 318:

“Co więcej, znamienny statystycznie wynik testu Kruskala-Wallisa czy ANOVA (p<0,05) tak naprawdę nie definiuje, między którymi grupami występuje istotność statystyczna. Aby rozwikłać ten problem, należy wykorzystać tzw. analizę post hoc z wykorzystaniem testów Tukeya czy najmniejszych istotnych różnic (NIR), które – jak test t – testują zmienne parami [11, 19].”

Translation into English:

“Furthermore, a statistically significant result of the Kruskal-Wallis test or ANOVA (p<0.05) does not really indicate between which groups there is statistical significance. To solve this problem, the so-called post-hoc analysis with Tukey’s tests or least significant differences (LSD) should be used, which – like the t-test – tests variables in pairs [11, 19].”

Thus, according to Krzych 2007, after the null hypotheses have been rejected by ANOVA or Kruskal-Wallis test, an appropriate post-hoc test for multiple comparisons between groups should be used, not the Student's t-test or the Mann-Whitney U-test (lines 511-515; Tables 1 and 2, in the revised manuscript). Using Student's t-test or Mann-Whitney U-test for multiple comparisons increases the rate of type I error (false positive), i.e. rejection of the null hypothesis when it is actually true. Although there are only three groups, appropriate post-hoc tests should be used, e.g. Tukey’s HSD and Dunn’s test, respectively.

The choice of post-hoc test for the two-way ANOVA is also problematic. The authors opted for Fisher’s LSD test (referred to as “Fisher’s NIR test” in the authors’ response, where NIR is the abbreviation for najmniejszych istotnych różnic, which means "least significant difference", i.e. LSD). The reason why Fisher’s LSD test should not be used in this case is the following: “Fisher’s LSD procedure is known to preserve the experimentwise type I error rate at the nominal level of significance, if (and only if) the number of treatment groups is three” (Meier U. A note on the power of Fisher's least significant difference procedure. Pharm Stat. 2006 Oct-Dec;5(4):253-63. doi: 10.1002/pst.210. PMID: 17128424). The larger the number of groups, the higher the type I error rate. In this experiment, the results in 6 (or 9, see my next comment) groups were compared (Figures 1 and 2, Tables A1 and A3). Since 6>3 (or 9>3), a suitable post-hoc test for multiple pairwise comparisons should be used, for example Tukey’s HSD.

4.2.    From the way the results are presented in Table 3 and in Tables A2 and A4, I was able to conclude that the authors examined the effects of exercise and body composition on the selected parameters by applying two two-way RM ANOVAs. Namely, for the values of the studied parameters in plasma with and without correction %dPV. So in the first ANOVA there were these 6 groups: NFAT-NLBM T0, NFAT-NLBM T1, NFAT-HLBM T0, NFAT-HLBM T1, HFAT-NLBM TO, HFAT-NLBM T1; and in the second ANOVA these 6 groups: NFAT-NLBM T0, NFAT-NLBM T1PV, NFAT-HLBM T0, NFAT-HLBM T1PV, HFAT-NLBM TO, HFAT-NLBM T1PV. Am I right? Figures 1 and 2 show all 9 groups and the p-values of the post-hoc test. Do these post-hoc test results come from two ANOVAs (with 6 groups each) or was one ANOVA performed for all 9 groups with one post-hoc test? This should also be explained in the manuscript (Materials and Methods, maybe also in Results or figure legends) so that readers do not have to guess.

5.       Results

5.1.    How are the data in Tables 1 and 2 presented and section 2.2? Mean±SEM? Mean±SD? It is not stated anywhere.

5.2.    Authors should report the values of the F and H statistics wherever they are missing (Tables 1 and 2 and in section 2.2, lines 140-141), as the p-values alone do not provide information about the size of a difference.

5.3.    I would suggest that the statistically significant values in Table 3 be bolded, as these are the most important results of this study. Also, put the units for AOPP to SOD in brackets, as for visfatin to irisin.

5.4.    Figures and tables should be self-sufficient. In Figures 1 and 2, some p-values are given, but the corresponding figure legends do not mention to which test these p-values refer. Tables A2 and A4 also show “post-hoc p-values” without specifying which post-hot test is involved. It is stated in Materials and Methods but it should be repeated in figures and tables.

5.5.    Lines 170-171: “Additionally, a significant increase in visfatin concentration was observed in the NFAT-NLBM (p<0.01) and NFAT-HLBM (p=0.04) groups…” Visfatin was only significantly increased in NFAT-HLBM without correction for %dPV.

Round 3

Reviewer 2 Report

Comments and Suggestions for Authors

In my opinion, this manuscript should be accepted for publication. I have a few minor comments.

Appendix tables A2 and A4. A total of five times in these two tables it writes 0.01. Is the p-value exactly 0.01 or <0.01 in these five cases?

Figure 1b. The boxes for T1 are gray, while these boxes are striped in all other graphs.

Lines 285-287. This sentence need linguistic revision. The intended meaning is not clear.

Correlations. I did not want to write about this in the previous reports so as not to distract from more important things. The authors used Pearson's correlation coefficient for data with a normal distribution and Spearman's for data that deviate from a normal distribution. This in itself is not a problem. At this point, I would like to point out that the Pearson correlation assesses linear relationships, while the Spearman correlation assesses monotonic relationships. In a linear relationship, the variables move in the same direction at a constant rate. In a monotonic relationship, the variables move in the same relative direction, but not necessarily at a constant rate. If the relationship is monotonic but not linear, the Spearman correlation coefficient is higher than the Pearson coefficient. Since I do not have the original data and scatter plots, I do not know what the relationships between the analyzed data are. However, the authors may, if they find it is justified, calculate the Spearman correlation coefficient for all pairs of data analyzed with the Pearson correlation coefficient to provide additional information about these relationships.
